# Winter cancellations of elective surgical procedures in the UK: a questionnaire survey of patients on the economic and psychological impact

Philip J J Herrod,[1] Alfred Adiamah,[2,3] Hannah Boyd-Carson,[1] Prita Daliya,[2] Ahmed M El-Sharkawy,[4] Panchali B Sarmah,[5] Tanvir Hossain,[2] Jennifer Couch,[2] Tanvir S Sian,[1] Andrew Wragg,[2] David R Andrew,[3] Simon L Parsons,[2] Dileep N Lobo,[2,6] WES-Pi Study Group on behalf of the East Midlands Surgical Academic Network (EMSAN)

PJJH and AA are joint first authors.

For numbered affiliations see end of article.

**Correspondence to**
Professor Dileep N Lobo;
dileep.lobo@nottingham.ac.uk

## ABSTRACT

**Objectives** To quantify the economic and psychological impact of the cancellation of operations due to winter pressures on patients, their families and the economy.

**Design** This questionnaire study was designed with the help of patient groups. Data were collected on the economic and financial burden of cancellations. Emotions were also quantified on a 5-point Likert scale.

**Setting** Five NHS Hospital Trusts in the East Midlands region of England.

**Participants** We identified 796 participants who had their elective operations cancelled between 1 November 2017 and 31 March 2018 and received responses from 339 (43%) participants.

**Interventions** Participants were posted a modified version of a validated quality of life questionnaire with a prepaid return envelope.

**Main outcome measures** The primary outcome measures were the financial and psychological impact of the cancellation of elective surgery on patients and their families.

**Results** Of the 339 respondents, 163 (48%) were aged <65 years, with 111 (68%) being in employment. Sixty-six (19%) participants had their operations cancelled on the day. Only 69 (62%) of working adults were able to return to work during the time scheduled for their operation, with a mean loss of 5 working days (SD 10). Additional working days were lost subsequently by 60 (54%) participants (mean 7 days (SD 10)). Family members of 111 (33%) participants required additional time off work (mean 5 days (SD 7)). Over 30% of participants reported extreme levels of sadness, disappointment, anger, frustration and stress. At least moderate concern about continued symptoms was reported by 234 (70%) participants, and 193 (59%) participants reported at least moderate concern about their deteriorating condition.

**Conclusions** The cancellation of elective surgery during the winter had an adverse impact on patients and the economy, including days of work lost and health-related anxiety. We recommend better planning, and provision of more notice and better support to patients.

### Strengths and limitations of this study

► This study presents aggregated evidence to support previous anecdotes of the importance of cancellation of elective operations to patients' well-being. It was devised with active consultation with patient groups. The questions asked were those that the patient groups thought relevant to the problem.

► It highlights the importance of early notification of cancellations in times such as winter pressures, where cancellations are unavoidable to sustain the delivery of emergency care.

► This study was retrospective by design and was potentially limited by recall bias.

► We achieved a less than 50% responder rate, in spite of two rounds of questionnaires being sent, and as such is subject to non-responder bias.

► The study covered a single but large region of the UK, it is difficult to ascertain if the population dynamics and level of social deprivation make this population similar to those in all parts of the UK, and as such affects the generalisability of our results.

## INTRODUCTION

The winter of 2017–2018 saw an unsustainable increase in patient demand on acute hospital care in England, due both to an anticipated seasonal rise in admissions and an unexpectedly severe epidemic of winter influenza. In response to what was dubbed the 'Winter National Health Service (NHS) Crisis' by the media, NHS England took the unprecedented step of instructing all NHS hospital trusts to cancel planned routine non-cancer surgery throughout December 2017 and January 2018, in order to free resources for emergency admissions.[1]

Although official figures have not yet been released, media estimates put the number

of patients affected in England at around 50 000.[2] This huge volume of cancellations will have ongoing ramifications for the 2018 calendar year.[3 4] The rescheduling of a large backlog of cancelled operations by hospital trusts, will undoubtedly put added pressure on hospital waiting lists.[3 4]

Despite the effect of a cancellation on the ongoing care provided by hospitals being subject to many reporting structures, the psychological and economic impact of a cancellation on patients themselves has received remarkably little attention. The requirement of a period of recuperation means most patients in their preparation for elective surgery make arrangements in their social lives to facilitate this. Adults in employment may book time off work and postpone important social events such as holidays. Carers may similarly organise additional support or childcare arrangements. Family members may also do the same in order to help care for recovering patients. If a planned operation is postponed at short notice, it is possible to envisage these arrangements resulting in personal and economic loss.

Any rescheduled operation date may also result in patients and carers incurring the same costs again with no recourse to reimbursement. This may lead to a considerable economic burden on the patient and their family[5] and cause additional loss of working days for the economy. Postponement or cancellation of elective surgery may also have a profound psychological impact on patients and their families, who may have mentally prepared themselves for the event. There may also be significant health-related anxiety associated with a longer wait for surgery, as patients may fear their condition deteriorating further.[6] We aimed, in this Winter Elective Surgery Cancellation and Psychological impact (WES-Pi) survey, to quantify the financial and psychological impact on patients following the postponement or cancellation of elective surgery in England, due to winter pressures.

## METHODS

This multicentre study was conducted across five NHS acute hospital trusts (Nottingham University Hospitals NHS Trust, University Hospitals of Leicester NHS Trust, Derby Teaching Hospitals NHS Foundation Trust, United Lincolnshire Hospitals NHS Trust and Chesterfield Royal Hospitals NHS Foundation Trust) in the East Midlands region of England during the winter of 2017–2018 in accordance with the STROBE statement.[7] Adult patients aged 16 years or over, scheduled for elective surgery between 1 November 2017 and 31 March 2018 were identified from local hospital waiting list records. As elective paediatric surgery is centralised and the majority of NHS Trusts contributing to this study do not provide a service, our focus was on adult patients only. Participants were chosen from the surgical subspecialties of general, colorectal, hepatopancreaticobiliary, upper gastrointestinal, endocrine, vascular, urology, orthopaedic, maxillofacial, and ear, nose and throat surgery. All patients who had their elective surgery postponed or cancelled for non-patient related reasons were included (figure 1). Patients who had surgery postponed or cancelled for patient-related reasons (eg, unrelated illness, medically unfit or surgery declined), or those that died before dispatch of the questionnaires were not invited to participate.

### Outcomes

Our primary outcome measures were to quantify the financial and psychological impact of cancellation of elective surgery on participants and their families. The secondary outcome measure was to evaluate the impact of the length of notice of cancellation on the primary outcomes.

### Survey design

In the absence of an existing validated questionnaire to explore financial or emotional consequences to a patient, specifically related to the cancellation of elective surgery, we modified a previously used survey from a paediatric

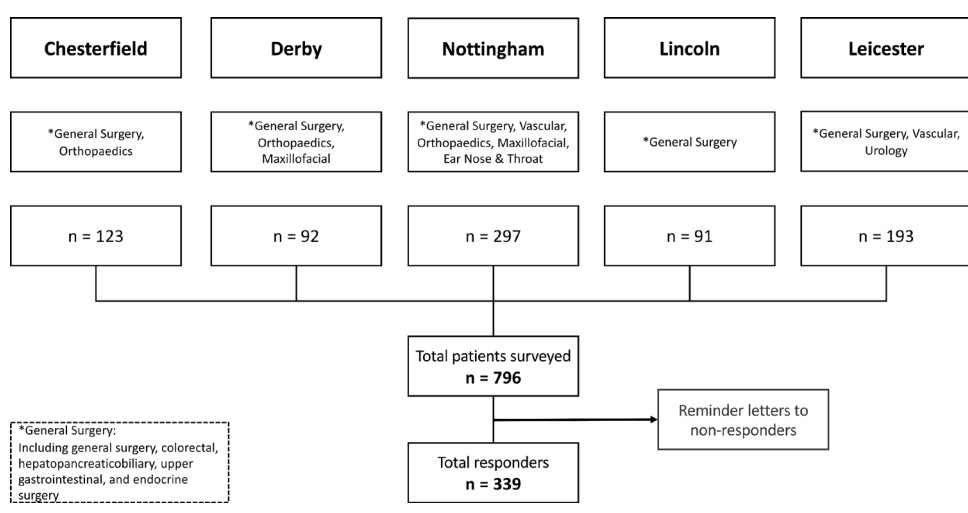

**Figure 1** Study flow diagram.

study.[8] Survey design involved multiple iterations and amendments through patient focus groups, with significant involvement and input from the Patient Advisory Group at Nottingham University Hospitals NHS trust, and the Patient Advisory and Liaison Service (PALS) team at Derby Teaching Hospitals NHS Trust. In addition to gathering data on the economic and financial burden of postponements and cancellations on participants and their relatives, the questionnaire also quantified emotional feelings of participants on a 5-point Likert scale[9] (online supplementary appendix 1).

## Patient and public involvement

Patient and public involvement (PPI) was an integral element throughout this study given its direct relevance to patient care. Embedding PPI into the study from the outset meant we could more accurately explore the patient experience of cancellations informed by what patients felt from prior experience were important in their care. There is a current drive for PPI in all medical research, for experiential reasons (patients or carers have experience of a condition, treatment or in this instance experience of a cancelled appointment for elective operation); ethical and moral reasons (patients and the public are directly affected by the research process and findings; they have the right to contribute to it; and of course are the tax payers); and pragmatic reasons (PPI is a requirement for funding bodies; it translates research into knowledge). In this instance, we felt strongly that for a study where the purpose was solely about patient experience, it had to be informed fully by patients.

We chose to recruit PPI lay assessors to review the invitation letter and subsequent survey to be sent to study participants. We wanted to prove that the intent to use a survey to explore the impact of cancelled elective operations on the psychosocial well-being of our patients, as a means leading to service improvement, if suitably structured would achieve its outcomes. PPI provided assurance that the devised series of questions contained in the survey was suitable to cover the aspects of financial, psychological and physical impact of these cancellations including symptomatic (suggestive) deterioration without adversely impacting on patients or being too onerous to complete.

They directed the changes through several iterations and approved of the final survey and invitation letter, ensuring the overall suitability and acceptability of the survey.

## How was the development of the research question and outcome measures informed by patients' priorities, experience and preferences?

In consultation with PPI members, the research tool was shaped to assess the financial impact of cancellations more accurately, not only for patients but also for their family members involved in their care. In exploring patients' emotions, their responses allowed us to make the questions more specific to feelings at the time of cancellations (online supplementary appendix 1).

## How did you involve patients in the design of this study?

From the PPI perspectives, the measures were informed by six PPI members recruited from the Gastrointestinal & Liver Patient Advisory Group, charged with reviewing the public-facing study documents (invitation letter and survey to be issued to the study participants). Each lay PPI member assessed and reported back their respective thoughts on the document(s) content, layout and its message. Examples of comments (on the survey) made and responses made by study leads are presented in online supplementary appendix 2.

## Were patients involved in the recruitment to and conduct of the study?

No. recruitment was informed by the cancellation list at each participating trust.

## How will the results be disseminated to study participants?

The write-up of the manuscript included the lead person from the gastrointestinal and liver PPI group, allowing the patient viewpoints to be reflected more accurately in our results. It also ensures dissemination to the initial group who were involved in the study design. More importantly there is an opportunity for this study to aid advocacy in the conduct of patient cancellations in a manner that results in the least adverse outcomes for patients.

## For randomised controlled trials, was the burden of the intervention assessed by patients themselves?

Not applicable.

## Ethics and Data collection

Identified participants were posted a copy of the study questionnaire (online supplementary appendix 1), a prepaid return envelope and an invitation letter (online supplementary appendix 2), incorporating local trust branding. Invitations were first posted to participants in April 2018, with a further invitation to non-responders in June 2018. Each study site was managed by local teams responsible for local study registration and approvals, identification of participants, dissemination of postal invitations and data collection. Returned surveys were pseudoanonymised, with collated data subsequently anonymised and encrypted before central data analysis. No identifiable participant data were transferred between sites.

## Data synthesis and analysis

Quantitative and qualitative data analyses of responses were undertaken. Quantitative data analysis was performed using STATA V.15.0 (StataCorp, College Station, Texas, USA). Linear regression was performed to assess for correlation and a chi-square test to assess for differences between categorical data. All descriptive data are represented as n (%), mean (SD) or median (IQR) as appropriate.

Qualitative data were evaluated using NVivo V.11 (QSR International, Burlington, Massachusetts, USA). For the purposes of thematic analysis, all the free text responses

**Table 1** Characteristics of the participants

| Characteristic | | n | % |
|---|---|---|---|
| Age | <65 years | 163 | 48.1 |
| | ≥65 years | 176 | 51.9 |
| Employment status | Employed | 98 | 28.9 |
| | Self-employed | 23 | 6.8 |
| | Retired | 192 | 56.6 |
| | Unemployed | 26 | 7.7 |
| Previous cancellation history | No cancellations | 72 | 25.0 |
| | 1 Cancellation | 106 | 36.8 |
| | 2 Cancellations | 65 | 22.6 |
| | ≥3 Cancellations | 45 | 15.6 |
| Operations | General* and vascular | 163 | 48.1 |
| | Orthopaedic | 135 | 39.8 |
| | Others† | 41 | 12.1 |
| Returns per hospital | Nottingham | 157 | 46.3 |
| | Derby | 42 | 12.4 |
| | Chesterfield | 42 | 12.4 |
| | Lincoln | 45 | 13.3 |
| | Leicester | 53 | 15.6 |

*Including general surgery, colorectal, hepatopancreaticobiliary, upper gastrointestinal and endocrine surgery.
†Urology, ear, nose, and throat surgery and maxillofacial surgery.

were transcribed and data validated by two study authors independently. These were then coded and analysed using the NVivo software into eight distinct themes that ranged from personal emotional response such as stress and anxiety; financial impact including cancelled holidays and time of work; and disease-related concerns such as worsening of symptoms. The number of responses within each theme were then quantified and exported into Microsoft Excel (Microsoft Corp., Redmond, Washington, USA) and presented visually as a radar diagram.

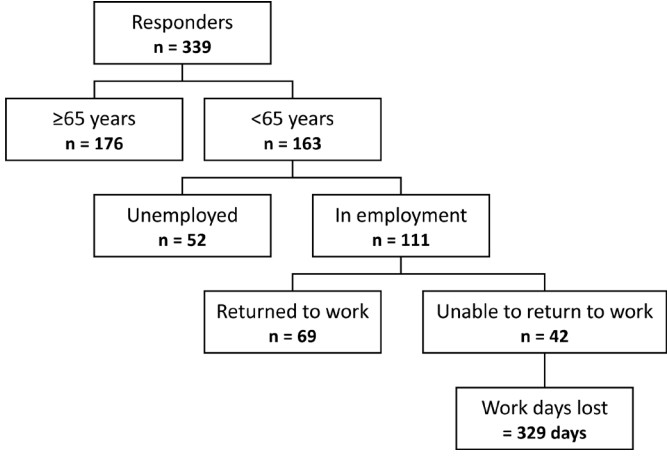

**Figure 2** Number of work days lost.

## RESULTS
### Return rate and cancellation
Questionnaires were posted to 796 participants and returned by 339 (43%) participants after two distribution rounds. The response rate varied between trusts from 27% to 52%. Data on participant characteristics and surgical specialties involved are displayed in table 1.

Of the participants returning questionnaires, 66 (19%) had their operations cancelled on the day of surgery, 149 (44%) within 24 hours of the proposed surgery and 125 (37%) cancelled more than 48 hours from the proposed date of the operation. Median notification of cancellation was 2 days (IQR 1–6). A number of participants had had their operations cancelled previously prior to the winter cancellation. The responders reported their operation as having been cancelled a median of one time (IQR 1–2) previously. In 15% (45/288) of the responders, operations had been cancelled three times or more (table 1).

### Economic burden
One hundred and sixty-three (48%) of responding participants were aged <65 years and of these 111 (68%) were in employment. Only 69 (62%) of the working adults were able to return to work during the time they had originally scheduled off for their operation, with a median loss of 1 day (IQR 0–5) of work. The total number of work days lost in the cohort was 329 (figure 2).

Additional working days in subsequent months were lost by 60 (54%) working adults due to the cancellation (median 3 days (IQR 0–12)), totalling 456 days. Family members of 111 (33%) participants required additional time off work (median 3 days (IQR 1–5)), totalling 581 days. Thus, the total number of missed days of work (from participants and family members, both from the initial cancellation and subsequent rescheduling of the operation) was 1366 days. There was no direct correlation between days of work lost and the notification given for cancellation ($r^2$=0.0004, p=0.87).

### Child care costs and travel costs
Of the responders with children, 23 were forced to organise extra childcare. Of these the median (IQR) additional childcare days required was 4 days (0–22.5). One hundred and twenty-three participants (48%) incurred additional travel costs following cancellation of their operation. The median cost incurred for additional travel was £15 (IQR £6.5–£30).

### Psychological burden
The majority of participants reported at least moderate levels of sadness, disappointment, anger, frustration and stress, with over 30% (n=102) of respondents reporting extreme levels of each of these emotions and over 50% of respondents reporting at least 'very' (the second highest answer available) to each of these emotions. A full breakdown of emotions reported is displayed in table 2. There was no significant difference in the reporting of these emotions between patients who had been cancelled only

| Table 2 | Responses to questions on psychological impact | | | | | | |
| --- | --- | --- | --- | --- | --- | --- | --- |
| | **Extremely** | **Very** | **Moderately** | **Slightly** | **Not at all** | **Not applicable** | **P value** |
| How sad were you? (Total n (%)) | 107 (32) | 56 (17) | 45 (13) | 70 (21) | 40 (12) | 21 (6) | |
| 1 Cancellation (n (%)) | 54 (30) | 27 (15) | 28 (15) | 41 (23) | 21 (12) | 10 (6) | 0.63 |
| >1 Cancellation (n (%)) | 53 (34) | 29 (18) | 17 (11) | 29 (18) | 19 (12) | 11 (7) | |
| How disappointed were you? (Total n (%)) | 150 (45) | 79 (24) | 37 (11) | 35 (10) | 28 (8) | 6 (2) | |
| 1 Cancellation (n (%)) | 72 (41) | 46 (26) | 20 (11) | 19 (11) | 15 (9) | 4 (2) | 0.73 |
| >1 Cancellation (n (%)) | 78 (49) | 33 (21) | 17(11) | 16(10) | 13(8) | 2 (1) | |
| How angry were you? (Total n (%)) | 107 (32) | 41 (12) | 45 (13) | 47 (14) | 88 (26) | 10 (3) | |
| 1 Cancellation (n (%)) | 51 (29) | 24 (13) | 25 (14) | 24 (13) | 50 (28) | 4 (2) | 0.69 |
| >1 Cancellation (n (%)) | 56 (35) | 17 (11) | 20 (13) | 23 (14) | 38 (24) | 6 (4) | |
| How frustrated were you? (Total n (%)) | 139 (41) | 58 (17) | 42 (12) | 52 (15) | 36 (11) | 12 (4) | |
| 1 Cancellation (n (%)) | 70 (39) | 36 (20) | 20 (11) | 30 (17) | 19 (11) | 5 (3) | 0.57 |
| >1 Cancellation (n (%)) | 69 (43) | 22 (14) | 22 (14) | 22 (14) | 17 (11) | 7 (4) | |
| How stressed were you? (Total n (%)) | 129 (38) | 49 (14) | 41 (12) | 42 (12) | 71 (21) | 7 (2) | |
| 1 Cancellation (n (%)) | 67 (36) | 28 (15) | 22 (12) | 26 (14) | 41 (22) | 2 (1) | 0.61 |
| >1 Cancellation (n (%)) | 62 (41) | 21 (14) | 19 (12) | 16 (10) | 30 (20) | 5 (3) | |

once or those who had suffered multiple cancellations. At least moderate concern about continued symptoms was reported by 234 (70%) patients while waiting for a rescheduled date while 193 (59%) reported at least moderate concern about their deteriorating condition.

There was no correlation between strength of any emotion and the amount of notice given to participants for the cancellation. This was true for all the emotive questions asked.

### Thematic analysis from free text responses

The qualitative analysis of the participants' free text responses is presented in the thematic analysis as a radar diagram (figure 3). Four particular themes were

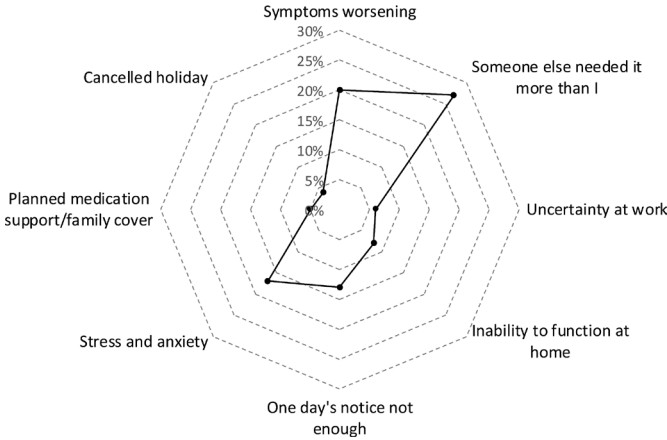

**Figure 3** Radar diagram of thematic analysis (figures represent percentage of responders).

highlighted in this analysis. There was concern over symptoms worsening; stress and anxiety related to the cancellation and those whose operations were cancelled on the day felt that this was unsatisfactory and caused them undue distress. However, the most interesting finding on the free text analysis was that 26% (70/268) of the responders understood the need for their operations to be cancelled to facilitate delivery of the emergency service. They used such descriptions as: "my operation was not life-threatening" and "others needed the care more" (box 1).

### DISCUSSION

The uncharacteristically cold weather, the high rates of emergency admissions, particularly in the elderly, and a lack of capacity in an already overworked and overstretched system gave rise to the perfect storm of cancellations during the winter of November 2017 to February 2018. Some even referred to it as another winter of discontent for the NHS.[10] The national strategy, instigated by NHS England was to cancel elective operations to release capacity in the system.[1] These elective cancellations were well publicised in the media, and meant that the majority of the population had some idea of the unusual stress the NHS was under.[10 11] However, despite public awareness, the impact on patients of these cancellations and, in particular last minute cancellations, was significant.

## Box 1 Sample of free text comments from responders

"I was aware of winter pressures and was expecting op to be cancelled. Felt it was right someone with greater need took priority."

"As I am on my own, with no family nearby, it had no real impact. Just a few phone calls to cancel lifts in and back. New appointment for 4th December, so not long to wait. Quite understood situation with the start of the "winter strain", so no point in getting stressed. No complaints at all."

"Struggling to swallow, risk of choking when eating."

"The surgeon was ready to do my operation on the correct day but was told at the last minute there was no bed for me in the ICU/high dependency ward so he couldn't do it. I was very disappointed, but it was not the surgeon's fault."

"I totally understood why my operation was cancelled. My operation needed to be done but my life was not in danger. It was just keeping my work informed so cover could be found whilst I was off work"

"It was cancelled the day before. This meant I had to re-arrange work, so did my partner and the childcare for 2 days. All at late notice. We all work full time. I was also due to fly to America for work and this now became an issue."

"I was disappointed on the original date being cancelled and postponed by 9/10 weeks. It was not a very serious life threatening situation for me and there were other serious cancer operations that required operations so I fully understood the reasons etc."

"Obviously at the time I felt sad because you not only gear yourself up to have the operation, but I could not justify taking up a hospital bed when there were more needy cases than mine requiring treatment. I do feel more money and less stress for NHS workers is required and the government needs to take heed!"

"I was waiting 6 months for this operation. This meant our lives were "put on hold" for this time. We could not book holidays, arrange anything that involved being away from home as we were expecting op dates; no information on delay times was given. This is very stressful"

"I was surprised to be given pre-med drugs before being told the operation was cancelled. I left the hospital feeling "spaced out"!"

"Each time I came for pre op washout the day before to be told when called in that there is no operation tomorrow. I have a young family so I have to prepare them emotionally."

"I am usually active and my hobbies and enjoyment usually is by way of playing sport. Not being able to undertake these activities had the obvious physical affects and resulted in me being in a low mood, even feeling depressed at times. In turn, I found myself comfort eating and drinking more alcohol. In conclusion, the physical and mental effects were profound."

"Operation performed 5 weeks later, did not affect outcome."

"Cancelled holiday. Developed sepsis after being discharged."

"Very disappointed."

"Such late notice when each operation was cancelled (day before). Very mentally draining preparing for our op then it getting cancelled."

"I was given a date. When I reached hospital there was no bed. Cancellation happened three times."

"My operation was cancelled due to a trauma case on that day. Everyone at the hospital was so kind and op was arranged for the next week. Thank you."

"My biggest problem has been the effect the original cancellation had on my elderly mother (aged 97) who was booked into respite care for six weeks @ £500 per week. She is still there some six months later. Her chances of coming home have diminished to almost none."

"It was rubbish at the time but completely understandable. Government should take the hit not the NHS. You guys are great and do all you can!"

Continued

## Box 1 Continued

"I was very distressed, I cried frequently and was very anxious. I felt let down and became depressed. I have not felt like doing anything and have rarely left the house. My irritable bowel syndrome got worse and my mental health has not been good and I have to take medication. My relationships have become strained and I feel I can't trust people."

"It made me feel that I didn't really matter! Although my condition is not life threatening, it is life debilitating."

### What this study found

This study is the first in England to evaluate the impact of the mass cancellation of elective surgery on patients, from both an economic and psychological view point. We have demonstrated that a large proportion of patients suffered a negative economic impact from both the additional work days lost and the additional non-refundable travel and childcare costs. We achieved this using a novel research tool, developed jointly with patient groups to establish usability, acceptability and internal validity.

Taken in the context of a nation-wide bed pressure crisis, which was widely publicised in the media, the fact that nearly 20% of patients had their operations cancelled on the day of surgery was surprising. Despite this we were unable to demonstrate any correlation between the period of notice given for cancellation of the planned operation, and the number of working days lost, nor any impact on the responses that patients gave to the questions on their emotional state. Although the reasons behind this are unclear, our study may have been underpowered to demonstrate this and also due to the over representation of retired patients in our cohort.

Our relatively small sample from one region of the UK, comprising five hospitals, contained nearly 800 cancelled operations and in those responders led to the loss of 1366 work days to the economy from our modest 43% response rate. Given that this crisis stretched to the whole length of the UK and was estimated to have led to 50 000 cancellations nationwide, the projected loss of working days may exceed 85 000 days, even on the unlikely assumption that the non-responding patients lost no days from work. The questions on psychological impact, demonstrated that elective surgical cancellation has led to a large amount of negative emotion among patients, with the majority displaying at least a moderate amount of health-related anxiety.

Although this study was about winter cancellations of operations of patients from one region of England, the results are probably generalisable to the rest of the UK and also to other countries. Moreover, the results are not just applicable to winter cancellations, but could be applied to cancellations occurring at other times of the year as well.

### What is available in the literature?

While several studies have evaluated the causes of bed pressures and failures of hospital trusts to meet the emergency

department 4 hour performance target[11 12] and others have evaluated the reason for elective cancellations across a range of healthcare settings,[13 14] very little information is available on the psychosocial impact of cancellations of elective operations on patients. Others have reported on measures to reduce same day cancellations of elective operations, but have focused on other aspects, such as ensuring fitness for surgery, preventing the scheduling of the wrong operation or preventing incomplete preoperative assessments rather than addressing cancellations due to a lack of beds.[15] Others have counted the cost of such last minute cancellations to the hospital itself, from specialised equipment that might need to be brought in, and noted that cancellations within 24 hours impacted significantly on costs to the trust.[16] In this study we found that 19% of the patients had their operations cancelled within 24 hours. While this resulted in a financial impact on the patients, such late cancellations are also likely to increase costs for the hospital trust itself. From the point of view of optimal deployment of personnel and resources, better planning is needed for future crises. The Royal College of Surgeons of England has published guidance to help manage winter pressures and has acknowledged that it was better for patients to be told in advance of their cancellations, in agreement with the findings of this study.[17] Additionally, they acknowledged the consequences of such cancellations inevitably would include prolonged suffering of the patients affected.[17] Better forecast and long-term planning of population health needs by policy makers are the only ways to avert these winter cancellations from becoming a recurring theme in the delivery of healthcare in the UK.

## Limitations

Our study may be limited by its response rate, potential recall bias and non-responder bias. We took steps to maximise our response rate by sending prepaid return envelopes with each questionnaire, and by dispatching a second round of questionnaires to non-responders. However, despite these efforts we were unable to improve our response rate above 50%. Unfortunately, as we did not receive any response at all from our non-responders, we do know the reasons for the lack of response. We, therefore, cannot exclude that non-response bias has not led to an exaggeration of the proportion of patients report with negative economic or psychological impacts. However, we would contend, that even with the unlikely assumption that the non-responders did not experience any negative emotion or loss of work, the economic and psychological burden on this cohort was considerable. Another major limitation of our study was the lack of a pre-existing, validated survey tool. In order to mitigate against this, we based our survey design on a previously existing survey for another population, modified in conjunction with PPI involvement from two hospital trusts.

## CONCLUSIONS

The WES-Pi survey has provided evidence to support the often-cited anecdotes concerning the impact of cancelling elective operations on patients' finances and psychological well-being. The national awareness of cancellations meant that the majority of responding patients understood the need for these cancellations. However, with 20% of respondents having their operations cancelled on the day of the procedure better planning is needed in the future to reduce cancellations and to provide more notice and better support to patients who have their operations cancelled. This may require a national change in policy by the Department of Health, in collaboration with specialist clinical commissioning groups and local trusts.

**Author affiliations**
[1]Derby Teaching Hospitals NHS Foundation Trust, Royal Derby Hospital, Derby, UK
[2]Nottingham Digestive Diseases Centre, National Institute for Health Research (NIHR) Nottingham Biomedical Research Centre, Nottingham University Hospitals NHS Trust and University of Nottingham, Nottingham, UK
[3]United Lincolnshire Hospitals NHS Trust, Lincoln County Hospital, Lincoln, UK
[4]Chesterfield Royal Hospitals NHS Foundation Trust, Chesterfield, UK
[5]University Hospitals of Leicester NHS Trust, Leicester, UK
[6]MRC/ARUK Centre for Musculoskeletal Ageing Research, School of Life Sciences, University of Nottingham, Queen's Medical Centre, Nottingham, UK

**Collaborators** Jennifer Couch, Prita Daliya, Tanvir Hossain, Bethan Johnson, Amanda Koh, Anisa Kushairi, Dileep N Lobo, Simon L Parsons, Andrew Wragg (Nottingham University Hospitals NHS Trust); Alfred Adiamah, David R Andrew, Chris Lewis-Lloyd, Farah Roslan, Amari Thompson, Helen Wan (United Lincolnshire Hospitals NHS Trust); Ahmed M El-Sharkawy, Sita Kotecha, Kevin Sargen (Chesterfield Royal Hospitals NHS Foundation Trust); Hannah Boyd-Carson, Phillip J J Herrod, Jonathan N Lund, Alastair Morton, Tanvir S Sian (Derby Teaching Hospitals NHS Foundation Trust); Edmund Charles, Panchali B Sarmah, Baljit Singh (University Hospitals of Leicester NHS Trust).

**Contributors** Study design: PJJH, AA, HBC, PD, AME-S, PBS, TH, JC, TSS, DRA, AW, SLP, DNL. Development of questionnaire: AA, PJJH, PD, HBC, AW. Data collection: PJJH, AA, HBC, PD, AME-S, PBS, TH, JC, TSS, DRA, AW, SLP, DNL. Data analysis: PJJH, AA, HBC. Data interpretation: PJJH, AA, HBC, PD, DNL. Writing of Manuscript: PJJH, AA, PD, HBC, DNL. Creation of figures: AA, PD, DNL. Critical review of manuscript: DRA, AW, SLP, DNL. Final approval: PJJH, AA, HBC, PD, AME-S, PBS, TH, JC, TSS, DRA, AW, SLP, DNL.

**Funding** This work was supported by the Medical Research Council [grant number MR/K00414X/1]; and Arthritis Research UK [grant number 19891]. AA was funded by a National Institute for Health Research (NIHR) Academic Clinical Fellowship.

**Disclaimer** The funders had no role in the design or conduct of the work, or in the decision to publish. This paper presents independent research funded by the MRC, ARUK and the NIHR. The views expressed are those of the authors and not necessarily those of the MRC, ARUK, NHS, the NIHR or the Department of Health.

**Competing interests** None declared.

**Patient consent for publication** Not required.

**Ethics approval** The study was registered and approved locally at each participating hospital trust, under the supervision of a named consultant surgeon as a service improvement project. The project registration/approval numbers for each site were: Nottingham University Hospitals NHS Trust 18-050c, University Hospitals of Leicester NHS Trust 9339E, Derby Teaching Hospitals NHS Foundation Trust SB-Gen-2017/18-1025, United Lincolnshire Hospitals NHS Trust 1301 and Chesterfield Royal Hospitals NHS Foundation Trust WES-Pi 001.

**Provenance and peer review** Not commissioned; externally peer reviewed.

**Data availability statement** Data are available upon reasonable request.

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
