## [Reviewer comments · BMJ Open]

ARTICLE DETAILS

TITLE (PROVISIONAL)	Winter cancellations of elective surgical procedures in the United Kingdom: A questionnaire survey of patients on the economic and psychological impact
AUTHORS	Herrod, Philip; Adiamah, Alfred; Boyd-Carson, Hannah; Daliya, Prita; El-Sharkawy, Ahmed; Sarmah, Panchali; Hossain, Tanvir; Couch, Jennifer; Sian, Tanvir; Wragg, Andrew; Andrew, David; Parsons, Simon; Lobo, Dileep

VERSION 1 - REVIEW

REVIEWER	Dr FARHANUL HUDA ALL INDIA INSTITUTE OF MEDICAL SCIENCES RISHIKESH UTTARAKHAND INDIA
REVIEW RETURNED	17-Jan-2019

GENERAL COMMENTS	WELL DESIGNED STUDY. ONLY DRAWBACK IS THAT IT IS A RETROSPECTIVE STUDY, RECALL BIAS IS AN ISSUE HERE. LESS NUMBER OF RESPONDENTS IS ANOTHER CRITICAL POINT. AUTHORS ARE SUGGESTED TO DO A PROSPECTIVE STUDY.
---

REVIEWER	melaku desta Debre Markos University
REVIEW RETURNED	28-Jan-2019

GENERAL COMMENTS	Thank you for the opportunity to review this manuscript. This paper assesses winter cancellations of elective surgical procedures: A questionnaire survey of patients on the economic and psychological impact in the United Kingdom. I would like to congratulate the authors for their effort in evaluating these issues at five National Health Service East Midlands, England For this, globally; the paper is relevant for two reasons. First, the paper is in line with the aims and scope of BMJ open. Second, few studies have conducted in the area of interest. Despite, the paper is absolutely poorly presented. It needs a linguistic revision and many clarifications and modifications. Abstract - Please correct grammatical, punctuation and, preposition errors.
---

- In the objectives and main outcome part: the word cancellation needs preposition the before it “the cancellation”.
 - The setting part, the word cancelled is misspelled and replaced by canceled and the word questionnaire in the participants part needs preposition the or a and see all others in respectively such as cancellation
 - Please replace the following phrase the primary outcome measure was the financial and psychological impact of cancellation by “the primary outcome measures were the financial and psychological impact of cancellation”. Hence you have two outcomes.
 - Who were your study participants? Please clearly stated?
 - Did you believe that your finding is valid if the tool was prepared in the way of patient group?
 - In the conclusions section, please include your recommendations based on what you found.
- Background
- It seems written in haphazard manner and taking a note instead of research.
 - Please rewrite with appropriate citations through reviewing literatures overall the section
 - It is not possible to use one citation for one paragraph e.g. page 6 lines 7-19.
 - Please edit in coherent manner and clearly for the reader without any redundancy of words or prepositions. E.g. page 6 line 23 and 36 repeats the conjunction although.
 - Please state problem statement clearly in line with your objective.
 - Please avoid any information without any source. E.g. Page 6 line 25-32 “this huge volume of cancellations will have ongoing ramifications for the 2018 calendar year. The rescheduling of a large backlog of cancelled operations by hospital trusts will undoubtedly put added pressure on hospital waiting lists.
 - Generally, please edit extensively the background section, with correcting grammar errors from the global to national then local burden of cancellation then go winter cancellation if that is the case – then go the economic and psychological impact of cancellation of elective surgery, then you should shows the interventions, gaps and come to your problem statement.
- Methods
- You should be revised extensively based on the BMJ author’s guideline and in clearly manner e.g based on the following order with using as sub- sections, study design and setting, study participant, selection criteria, study variables and measurements, data collection and quality control, data analysis and, public involvement.
 - Why you are interested to include only adult patients aged 16 years over scheduled for elective surgery? is that pediatric age group were not cancelled for surgery ?
 - Please correct grammar and spelling errors and improper punctuations. E.g. page 8 line 7 needs preposition the before five hospitals, page 8 line 19 the word orthopaedics should be replaced by “orthopedics”, page 8 line 26 come (,) before or should removed.
 - Please avoid unnecessary words or sentence. E.g. page 8 line 29 those that died before dispatch of the questionnaires were not invited to participate, what does that mean? What is the importance of this sentence? Already those died were already not your participant.
 - How many you were approached, included in the study and have been retained.

	 - Reasons for non-participation should be reported clearly - Please validate the survey. Hence , page 8 line revealed that there is no validated questionnaire to explore financial or emotional consequences - When we say that the survey is validated? - In the patient and public involvement part, page 9 line 48 the spelling tax payers should replaced by taxpayers. - What type of analysis you undertake for qualitative data. - Please show any appropriate reporting statement guideline with citation and place as supplementary material e.g. STROBE or others. Results and Discussion  - Please avoid grammar errors similarly. - In results section, please reduce the sub-sections e.g the sub-sections children costs and travel costs in page 14 can be included under the economic burden. - Please justify your finding with others with others. Major revisions is recommended As mentioned before, the overall the manuscript is poorly presented. It needs extensive editing service before process for publication. We can find several typing errors in each page. Consequently, I'll not waste my time to note all these errors. This task has to be done by the authors before submitting their manuscript.
--	--

REVIEWER	Arnold Hill / Paul Healy Department of General Surgery, Beaumont Hospital and The Royal College of Surgeons in Ireland, Beaumont, Dublin 9, Ireland
REVIEW RETURNED	14-Mar-2019

GENERAL COMMENTS	The authors should be congratulated on organising and coordinating this multicentre study assessing the economic and psychological impact of procedure cancellations on patients and their families. It is worthy of study and places a human perspective on the impact of high volume cancellations on the individual patient. The introduction clearly identifies the scale of the problem of cancellations and the potential problems that may face patients and their families. Following this, the intended aim of the study has been clearly stated. The authors have not used a previously validated survey but have clearly described the design process of the survey used. It is at times quite protracted though. The response rate was low but this has been acknowledged by the authors as a limitation. It might have been beneficial to include an analysis of the characteristics of the non-responders from information available eg demographics, type of procedure canceled, number of times procedure canceled. Is there any explanation available at a Trust level to explain the short noticed many received when canceling their procedure? As stated in the introduction NHS England instructed Trusts to cancel planned routine noncancer surgery in December and January. Despite this, there was a significant number of cancellations for
--

	nonpatient reasons at just 5 hospitals at very short notice. An understanding of such factors may facilitate better planning and reduce the burden, financial and economic, on patients and their family. I appreciate that this is not an intended aim of this study. The number of days off of work for both patients and family who have had procedures canceled is significant. There is, however, no apparent explanation for such a dramatic figure of 1366 days for a small number of patients. Or is this an accumulative figure for multiple cancellations? Future studies/questionnaires may question in the IQR of days off reported. There is a balanced reporting of the economic and psychological consequences of the impact of cancellations on patients and their family. However (based on a rough visual calculation) there seems to be almost 50% of participants reporting to be very or extremely sad/disappointed/angry/ frustrated and stressed. This is significant and I believe should be referred to in the results sections of the abstract and main text "The fact that nearly 20% of patients had their operations canceled on the day of surgery was unacceptable" This is the authors' opinion (and rightly so in that it is unacceptable) but this can not be concluded from the current study results from what I can see. In fact, they acknowledge that there was no correlation in impact on responses between the period of notice given to cancel the procedure Figure 2- in employment total 111 but return to work 69 and unable to return to work 62 = 131 Table 2-Characteristics of the participants. A significant number of patients (>35%) had their procedure canceled on 2 or more occasions. I would like to see a subanalysis of the response to questions on psychological impact by the number of times patients had their procedures canceled
--	--

REVIEWER	Grant Cumming NHS Scotland
REVIEW RETURNED	18-May-2019

GENERAL COMMENTS	well written paper enjoyed reading this might have been worth including gynaecological operations as usually first to be cancelled as large proportion benign disease
---

VERSION 1 – AUTHOR RESPONSE

Reviewer 1

WELL DESIGNED STUDY.

ONLY DRAWBACK IS THAT IT IS A RETROSPECTIVE STUDY, RECALL BIAS IS AN ISSUE HERE.

LESS NUMBER OF RESPONDENTS IS ANOTHER CRITICAL POINT.

AUTHORS ARE SUGGESTED TO DO A PROSPECTIVE STUDY

We thank the reviewer for their praise of our study design. We have already listed the key limitations of our study that you have highlighted (response rate and potential recall bias) in the limitations section of our discussion, together with the steps we took to minimize these.

We agree with the reviewer for the need for a prospective study, which we believe could now be carried out, using the questionnaire we have designed and used in the present study.

Reviewer 2

Thank you for the opportunity to review this manuscript.

This paper assesses winter cancellations of elective surgical procedures: A questionnaire survey of patients on the economic and psychological impact in the United Kingdom. I would like to congratulate the authors for their effort in evaluating these issues at five National Health Service East Midlands, England. For this, globally; the paper is relevant for two reasons. First, the paper is in line with the aims and scope of BMJ open. Second, few studies have been conducted in the area of interest.

We thank the reviewer for their positive comments on the importance of our study.

Despite, the paper is absolutely poorly presented. It needs a linguistic revision and many clarifications and modifications.

Abstract

- Please correct grammatical, punctuation and, preposition errors.
- In the objectives and main outcome part: the word cancellation needs preposition the before it "the cancellation".
- The setting part, the word cancelled is misspelled and replaced by canceled and the word questionnaire in the participants part needs preposition the or a and see all others in respectively such as cancellation.

We have checked and have corrected the few grammatical and spelling errors.

- Please replace the following phrase the primary outcome measure was the financial and psychological impact of cancellation by "the primary outcome measures were the financial and psychological impact of cancellation". Hence you have two outcomes.

This has been corrected

- Who were your study participants? Please clearly state?

We state in the abstract that our participants were the 339 patients who had an elective operation cancelled between 1-11-2017 and 31-3-2018. These were patients scheduled for an operation at one of the 5 NHS trusts participating in this study,

- Did you believe that your finding is valid if the tool was prepared in the way of patient group?

The tool was modified from previous questionnaire studies. Yes, we think that Patient and public involvement in research design is vital and is actively encouraged by the BMJ open <https://blogs.bmj.com/bmjopen/2018/03/23/new-requirements-for-patient-and-public-involvement-statements-in-bmj-open/>

- In the conclusions section, please include your recommendations based on what you found.

We thank the reviewer for highlighting this and have added a line to the conclusions

Background

- It seems written in haphazard manner and taking a note instead of research.
- Please rewrite with appropriate citations through reviewing literatures overall the section

As the reviewer describes earlier in their comments, very few studies have been conducted in this area of research and feel that we have already cited the required literature. We have however made some additional references.

- It is not possible to use one citation for one paragraph e.g. page 6 lines 7-19.

We are unsure as to what the reviewer means by this comment

- Please edit in coherent manner and clearly for the reader without any redundancy of words or prepositions. E.g. page 6 line 23 and 36 repeats the conjunction although.

We have changed the repeated word although

- Please state problem statement clearly in line with your objective.

We have stated our aim at the end of our introduction- to quantify the financial and psychological impact on patients...

- Please avoid any information without any source. E.g. Page 6 line 25-32 "this huge volume of cancellations will have ongoing ramifications for the 2018 calendar year. The rescheduling of a large backlog of cancelled operations by hospital trusts will undoubtedly put added pressure on hospital waiting lists.

We would contend that this is the logical conclusion of our argument and does not require referencing. However we agree there is no published data for us to cite here.

Nevertheless, we have added 4 new references and the total number of references in the bibliography is now 15.

- Generally, please edit extensively the background section, with correcting grammar errors from the global to national then local burden of cancellation then go winter cancellation if that is the case – then go the economic and psychological impact of cancellation of elective surgery, then you should shows the interventions, gaps and come to your problem statement.

This has been done

Methods

- You should be revised extensively based on the BMJ author's guideline and in clearly manner e.g. based on the following order with using as sub- sections, study design and setting, study participant, selection criteria, study variables and measurements, data collection and quality control, data analysis and, public involvement.

The original submission was according to the journal guidelines. We leave the decision regarding this to editorial discretion.

- Why you are interested to include only adult patients aged 16 years over scheduled for elective surgery? is that pediatric age group were not cancelled for surgery ?

We did not include paediatric patients in our research, firstly as this would have required many additional ethical approvals and secondly the NHS winter crisis of 2018 was mainly in adult health care and led to a shortage of adult inpatient beds. With paediatric patients, comments would be from carers/parents. Hence we concentrated on adults for this study.

- Please correct grammar and spelling errors and improper punctuations. E.g. page 8 line 7 needs preposition the before five hospitals, page 8 line 19 the word orthopaedics should be replaced by "orthopedics", page 8 line 26 come (,) before or should removed.

We thank the reviewer for raising these issues, however we disagree with them. There are more than 5 acute hospitals in the East Midlands region, and more could have been included if we had had the logistical means. Thus we are unsure what preposition the reviewer would like us to use, as use of the preposition "the" would incorrectly lead the reader to assume there are only 5 hospitals in the region. "orthopedics" is not the correct spelling in UK English.

- Please avoid unnecessary words or sentence. E.g. page 8 line 29 those that died before dispatch of the questionnaires were not invited to participate, what does that mean? What is the importance of this sentence? Already those died were already not your participant.

The questionnaires were dispatched in April 2018, asking about cancellations that took place between November 2017 and March 2018. Thus, some patients who had suffered a cancellation unfortunately died before questionnaire dispatch and we took care not to send a questionnaire to their home address in order to avoid any distress this may have caused to family members.

- How many you were approached, included in the study and have been retained.

This is detailed in figure 1. We dispatched 796 questionnaires to eligible patients and received 339 responses.

- Reasons for non-participation should be reported clearly

The reason for non- participation in all cases was lack of response. Unfortunately, we do not know the reason why any of the patients failed to return their questionnaire. They did not return the questionnaire, despite a second invitation and we did not have ethical permission to contact them in other ways.

- Please validate the survey. Hence , page 8 line revealed that there is no validated questionnaire to explore financial or emotional consequences

- When we say that the survey is validated?

As we state in the manuscript, a validated survey does not exist, therefore we undertook to design our own, based on a previously published one in the literature.

- In the patient and public involvement part, page 9 line 48 the spelling tax payers should be replaced by taxpayers.

We do not believe that this would be correct

- What type of analysis you undertake for qualitative data.

Our qualitative data is first reported by descriptive statistics and now a subgroup analysis dividing patients by number of previous cancellations has been conducted in accordance with reviewer 3's comments. The qualitative analysis specifically relating to the patients "free text responses" were analysed using the Nvivo software. This allowed us to generate classification and themes based on the respondents free text. The 8-themes generated included examples such as "short notice of cancellations, fear of worsening symptoms, stress and anxiety." The proportions expressing each of these themes were then expressed graphically as a spider diagram (fig 3).

- Please show any appropriate reporting statement guideline with citation and place as supplementary material e.g. STROBE or others.

We have included a reference to STROBE and have completed the relevant checklist

Results and Discussion

- Please avoid grammar errors similarly.

- In results section, please reduce the sub-sections e.g the sub-sections children costs and travel costs in page 14 can be included under the economic burden.

We have combined the relevant sections

- Please justify your finding with others with others.

We have amended our discussion to do this and added an additional reference.

Major revisions is recommended

As mentioned before, the overall the manuscript is poorly presented. It needs extensive editing service before process for publication. We can find several typing errors in each page. Consequently, I'll not waste my time to note all these errors. This task has to be done by the authors before submitting their manuscript.

We thank the reviewer for their comments.

Reviewer: 3

Reviewer Name: Arnold Hill / Paul Healy

Institution and Country: Department of General Surgery, Beaumont Hospital and The Royal College of Surgeons in Ireland, Beaumont, Dublin 9, Ireland

Please state any competing interests or state 'None declared': None declared

Please leave your comments for the authors below

The authors should be congratulated on organising and coordinating this multicentre study assessing the economic and psychological impact of procedure cancellations on patients and their families. It is worthy of study and places a human perspective on the impact of high volume cancellations on the individual patient.

We thank the reviewer for these comments

The introduction clearly identifies the scale of the problem of cancellations and the potential problems that may face patients and their families. Following this, the intended aim of the study has been clearly stated.]

We thank the reviewer for these comments

The authors have not used a previously validated survey but have clearly described the design process of the survey used. It is at times quite protracted though.

This questionnaire was assessed and approved by a PPI group.

The response rate was low but this has been acknowledged by the authors as a limitation. It might have been beneficial to include an analysis of the characteristics of the non-responders from information available eg demographics, type of procedure canceled, number of times procedure canceled.

Unfortunately this is not possible as we were not granted access to this data from the hospital administrators. The demographics we report in table 1 are taken from the responses the patients gave in their questionnaires

Is there any explanation available at a Trust level to explain the short notice many received when canceling their procedure? As stated in the introduction NHS England instructed Trusts to cancel planned routine noncancer surgery in December and January. Despite this, there was a significant number of cancellations for nonpatient reasons at just 5 hospitals at very short notice. An understanding of such factors may facilitate better planning and reduce the burden, financial and economic, on patients and their family. I appreciate that this is not an intended aim of this study.

The main reason for this was lack of beds for elective operations due to a vast, seasonal increase in emergency admissions

The number of days off of work for both patients and family who have had procedures canceled is significant. There is, however, no apparent explanation for such a dramatic figure of 1366 days for a small number of patients. Or is this an accumulative figure for multiple cancellations? Future studies/questionnaires may question in the IQR of days off reported.

The figure of 1366 days comes as a cumulative total of the 329 days lost by the patients themselves directly as a result of the cancellation (i.e. the time they had scheduled off work for the operation they did not have, and were unable to go back into work for), the 456 days of work that the patients lost due to subsequent scheduling of their postponed operation or ill health, and the 581 days of work lost by family members. We have added a line to the manuscript to clarify this.

There is a balanced reporting of the economic and psychological consequences of the impact of cancellations on patients and their family. However (based on a rough visual calculation) there seems to be almost 50% of participants reporting to be very or extremely sad/disappointed/angry/ frustrated and stressed. This is significant and I believe should be referred to in the results sections of the abstract and main text

We agree with the reviewer on the importance of highlighting this and have done so in both the abstract and the results.

"The fact that nearly 20% of patients had their operations canceled on the day of surgery was unacceptable" This is the authors' opinion (and rightly so in that it is unacceptable) but this can not be concluded from the current study results from what I can see. In fact, they acknowledge that there was no correlation in impact on responses between the period of notice given to cancel the procedure

This has been modified

Figure 2- in employment total 111 but return to work 69 and unable to return to work 62 = 131

We apologise for this typing error, which has been corrected.

Table 2-Characteristics of the participants. A significant number of patients (>35%) had their procedure canceled on 2 or more occasions. I would like to see a subanalysis of the response to questions on psychological impact by the number of times patients had their procedures canceled

The reviewer is correct that a significant number of patients were cancelled multiple times. In order to attempt this analysis we have performed a subgroup analysis, splitting the cohort into those with 1 cancellation and those with more than one and have added this to table 2 (previously table 3). We have then performed a chi-square test, testing the hypothesis that those who were cancelled more often would be more adversely effected psychologically, however we have not found this to be the case.

Reviewer: 4

Reviewer Name: Grant Cumming

Institution and Country: NHS Scotland

Please state any competing interests or state 'None declared': none

Please leave your comments for the authors below

well written paper

enjoyed reading this

We thank the reviewer for these comments

might have been worth including gynaecological operations as usually first to be cancelled as large proportion benign disease

We agree with the reviewer, but the intention was not to include gynaecology for this survey. In several of the hospitals included in this study, the gynaecology department is contained within a separate organisational division and would have required additional permissions to include.